# Relationships between the hard and soft dimensions of the nose in *Pan troglodytes* and *Homo sapiens* reveal the positions of the nasal tips of Plio-Pleistocene hominids

Ryan M. Campbell[1]*, Gabriel Vinas[2], Maciej Henneberg[1,3]

1 Adelaide Medical School, Biological Anthropology and Comparative Anatomy Research Unit, The University of Adelaide, Adelaide, South Australia, Australia, 2 Sculpture Department, Herberger Institute for Design and the Arts, Arizona State University, Tempe, Arizona, United States of America, 3 Institute of Evolutionary Medicine, Faculty of Medicine, University of Zurich, Zurich, Switzerland

* ryan.campbell@adelaide.edu.au

**Data Availability Statement:** All relevant data are within the paper and its Supporting information files.

## Abstract

By identifying homogeneity in bone and soft tissue covariation patterns in living hominids, it is possible to produce facial approximation methods with interspecies compatibility. These methods may be useful for producing facial approximations of fossil hominids that are more realistic than currently possible. In this study, we conducted an interspecific comparison of the nasomaxillary region in chimpanzees and modern humans with the aim of producing a method for predicting the positions of the nasal tips of Plio-Pleistocene hominids. We addressed this aim by first collecting and performing regression analyses of linear and angular measurements of nasal cavity length and inclination in modern humans (*Homo sapiens*; $n = 72$) and chimpanzees (*Pan troglodytes*; $n = 19$), and then performing a set of out-of-group tests. The first test was performed on four subjects that belonged to the same genus as the training sample, i.e., *Homo* ($n = 2$) and *Pan* ($n = 2$), and the second test, which functioned as an interspecies compatibility test, was performed on *Pan paniscus* ($n = 1$), *Gorilla gorilla* ($n = 3$), *Pongo pygmaeus* ($n = 1$), *Pongo abelli* ($n = 1$), *Symphalangus syndactylus* ($n = 3$), and *Papio hamadryas* ($n = 3$). We identified statistically significant correlations in both humans and chimpanzees with slopes that displayed homogeneity of covariation. Prediction formulae combining these data were found to be compatible with humans and chimpanzees as well as all other African great apes, i.e., bonobos and gorillas. The main conclusion that can be drawn from this study is that our set of regression models for approximating the position of the nasal tip are homogenous among humans and African apes, and can thus be reasonably extended to ancestors leading to these clades.

## Introduction

The process of producing faces from dry skulls is known as facial approximation. Since the purpose of this procedure is to estimate the premortem anatomy of individuals as closely as

**Funding:** The author(s) received no specific funding for this work.

**Competing interests:** The authors have declared that no competing interests exist.

possible, each facial feature requires a robust scientific method. While each feature of the face is important in its own right, approximating the position of the nasal tip is critical because the nose is in the center of the face. Approximation error related to nasal anatomy can thus significantly change the facial appearance of the subject in question. For this reason, the tip of the nose (pronasale) is a prominent landmark in the forensic facial approximation literature [1–6].

The pronasale landmark is equally important in facial approximations of extinct Plio-Pleistocene hominids; in this paper hominids means all members of the Hominidae, which is comprised of the African apes, humans, and all ancestors leading to these clades. It has been stated that primate comparative anatomy, which is the study of similarities and differences in structures of different species, is critical to the practice of ancient hominid facial approximation [7]. However, despite numerous facial approximations of extinct hominids presented in scientific textbooks and museum displays, interspecific variation in soft tissue nasal form, or any other feature for that matter, between humans and chimpanzees has received little scientific interest. Although some overlap between human and chimpanzee noses is documented, why modern humans possess a particularly unique projecting, external nose is essentially a mystery. In contrast to human noses, the noses of chimpanzees, and of other great apes (bonobos, gorillas, and orangutans), are relatively flat. Therefore, an investigation into the morphological differences between extant hominids may result in more scientifically robust facial approximation methods, which are needed to reduce the excessive variability recognized in facial approximations of the same individual [8, 9].

For a full overview of the complexities involved in forensic facial approximation, see Stephan et al. [10] and references therein. Here, we discuss only those methods relevant to approximating the nasal profile. In the facial approximation literature, eight methods for approximating the nasal profile in modern humans have been published [1, 3–5, 11–14]. Studies testing these methods [1, 15] have consistently reported that the method by George [12] appears to be the most useful. It consists of calculating a percentage (60.5% for males and 56% for females) of a distance from nasion to the inferior nasal spine to establish a chord at subnasale parallel to the Frankfurt horizontal plane. More recently, Burton et al. [16] found the morphology of the nasal bridge useful for inferring the shape of the nasal tip with very high accuracy and repeatability. Furthermore, orthodontists and maxillofacial surgeons, who also share an interest in nasal morphology, have identified numerous correlations between the soft and hard nasal tissues [17]. These correlations were first explored in Stephan et al. [1], then in Rynn et al. [3], and most recently in Allan et al. [18]. All of these studies have produced regression equations for approximating nasal morphology from dry skulls, although the method by Stephan et al. [1] has been shown to underestimate nasal protrusion [15]. Regardless of the validity and reliability of these methods there are still no serious scientific studies supporting their use on fossil hominids.

Most evolutionary studies of the nasal region have focused on modern humans [19–21] and Neanderthals/Neandertals [22–26]. Conversely, very little attention has been paid to the soft tissue of great ape noses. While we acknowledge that chimpanzee noses have received some research interest [27–29], with the exception of one study [30], gorilla and orangutan studies are practically non-existent. Given that great apes have been useful for understanding the human evolutionary story, it is not clear why the nasal soft tissues of great apes are so understudied. However, it is likely a direct result of the status of these animals as endangered species and the current difficulties involved in obtaining this kind of material (pers. observation).

It has been said that among all the transformations in craniofacial morphology recognized to have occurred during the proposed—although contested (see Kimbel and Villmoare [31])—transition from *Australopithecus* to *Homo*, nasal morphology is a major one [32]. In particular, the nose of modern humans is distinguished from great apes by possessing an external part

that protrudes past the piriform aperture. The evolutionary reasons for this feature are the subject of continuing scientific debate. Some studies argue the external nose was derived in the genus *Homo* because of adaptation to climate [33–36]. According to one hypothesis [32], the nose arose out of the face through expansion of the nasal bones relative to the piriform aperture. This is said to have occurred as Pleistocene hominids, such as *Homo erectus*, shifted to increasingly prolonged bouts of physical activity in arid environments, resulting in selective pressures on adaptive respiratory function [32]. In contrast, a more recent study by Nishimaru et al. [37] showed that the external protruding nose in modern humans has little effect on improving air conditioning. They concluded that the unique nasal anatomy in *Homo* was likely formed passively by facial reorganization and not from adaptation to climate. It is important to note that neither of these explanations provide the empirical data needed to approximate the nasal profiles of Plio-Pleistocene hominids. In addition, the question of whether the protrusion growth of the nose was constant or punctuated is entirely unanswered. Obtaining this knowledge is crucial to inform practitioners of facial approximation of how to model the nasal anatomy of their subjects and produce more accurate facial approximations of hominids. Thus, further research is needed to compare the nasal region among extant hominids.

The aims of this study were to explore the matter of Plio-Pleistocene hominid facial approximation further with the focus on predicting the position of the nasal tip. We aimed to (1) compare pronasale position in modern humans and chimpanzees, and (2) to produce prediction formulae for approximating the nasal protrusions of ancient Plio-Pleistocene hominids. We addressed these aims twofold: Firstly, by collecting and performing regression analyses of linear and angular measurements of nasal cavity length and inclination in modern humans (*Homo sapiens*; n = 72) and chimpanzees (*Pan troglodytes*; *n* = 19); and secondly, by performing a set of out-of-group tests consisting of 16 individuals. The first test was performed on four subjects that belonged to the same genus as the training sample, i.e., *Homo* (*n* = 2) and *Pan* (*n* = 2), and the second test, which functioned as an interspecies compatibility test, was performed on *Pan paniscus* (*n* = 1), *Gorilla gorilla* (*n* = 3), *Pongo pygmaeus* (*n* = 1), *Pongo abelli* (*n* = 1), *Symphalangus syndactylus* (*n* = 3), and *Papio hamadryas* (*n* = 3). We hypothesize that soft tissue approximation models, such as those for nasal protrusion, are homogenous among extant hominids and can thus be reasonably extended to all ancestors leading to these clades. To illustrate this hypothesis, we approximated the nasal protrusions for nine fossil hominid specimens. Given the fragile nature of the bones that make up the nasal cavity and how this diminishes the likelihood of their preservation in fossil crania, it was decided to take the least number of measurements needed to produce the prediction formulae. It would simply make no sense, in the context of ancient hominid facial approximation, to produce formulae that require measurements of intricate structures, such as those of the conchae of ethmoid, of extant species if these measurements could not be collected from fossils due to a severely low probability of preservation. Therefore, we only took measurements from aspects of the skull base and maxillofacial skeleton that are most durable and best protected against taphonomic deformation.

## Materials & methods

The material used in this study consists of 19 computed tomography (CT) scans of chimpanzee (*P. troglodytes*) heads, previously analyzed in Campbell et al. [38], and 72 lateral cephalometric radiographs of humans. The chimpanzee sample was collected as Digital Imaging and Communications in Medicine (DICOM) format bitmap files from the Digital Morphology Museum, KUPRI (dmm.pri.kyoto-u.ac.jp). The sex ratio for the chimpanzee sample was 1:1.71 (7 male and 12 female) and the mean age 30.9 years (minimum = 9; maximum = 44;

SD = 10.1). A complete list of all the chimpanzee subjects used in this study is presented in the S1 Table. The human sample was collected from the archive of a previous study [39]. The human sample includes two populations from different geographic areas: a Chinese population ($N$ = 52), and an American/European population ($N$ = 20). The sex ratio for the Chinese sample was 1:0.79 (29 male and 23 female). Exact ages were not available for this group but are classified as young adult. The sex ratio for the American/European sample was 1:0.82 (11 male and 9 female) and the mean age was 19 years and 1 month (minimum = 15; maximum = 32; SD = 4.7). We chose not to investigate sex differences in our samples for the following reasons: First, sex of archaic hominids is often subject to debate in paleoanthropology and, therefore, there is no practical reason to have independent methods for approximating the nasal tip in males and females separately because these methods cannot be confidently assigned to hominids. This is especially the case for those earliest members of the genus *Australopithecus*, such as Sts 5 [40]; and second, the aim of our study is to investigate covariation in soft and hard tissue variables between separate species, not averages of those variables between individuals classified by sex. Ethical approval was not required for the use of human subjects in this study due to the archival and anonymous nature of this material.

Measurements were taken on the midsagittal plane from the chimpanzee DICOM files in OsiriX MD, v. 11.02 (Visage Imaging GmbH, San Diego, USA), and from physical copies of the human radiographs with sliding and spreading calipers. Linear distances were collected using four standard cephalometric landmarks: basion (ba), nasion (n), pronasale (pn), and prosthion (pr) (Fig 1). These four landmarks were positioned onto the skulls and then measurements were taken for cranial base length (ba-n), nasal cavity length (ba-pn), and basion-prosthion length (ba-pr; hereafter referred to as jaw protrusion). All of these measurements were taken according to their descriptions in Martin and Saller [41], which are listed in Table 1. These measurements were chosen because they are relevant to approximating pronasale for the following reasons: 1) cranial base length has been shown to be correlated with other dimensions of the body, such as body height [42] and jaw base size [43]. Cranial base length may therefore be correlated with other dimensions of the body, such as nasal cavity length; and 2) all three of these measurements are generally well represented in the hominid fossil record. As stated in the Introduction, there would simply be no point in taking measurements of intricate structures from extant human and great ape skulls if these measurements could not also be collected from extinct hominid skulls.

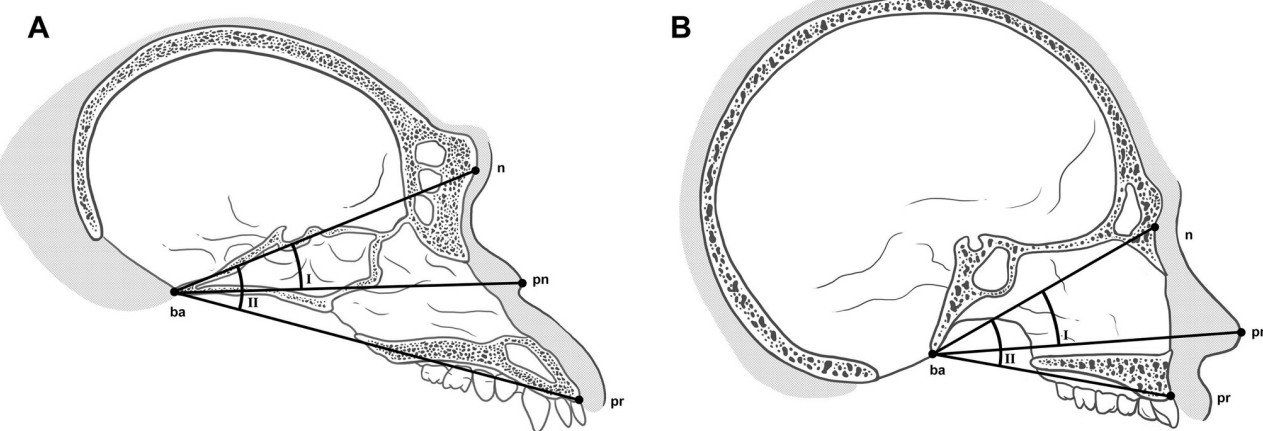

**Fig 1.** Locations of cephalometric landmarks used in this study and angles measured on the skull of a chimpanzee (*Pan troglodytes*; A) and modern human (*Homo sapiens*; B) in norma lateralis. (I) n-ba-pn angle. (II) n-ba-pr angle. See variable abbreviations in Table 1.

**Table 1. Cephalometric landmarks used in this study including their abbreviations and definitions.** Points are listed in alphabetical order for ease of reference.

| Landmark | Abbreviation | Definition |
|----------|--------------|------------|
| **Basion** | ba | Anteriormost point of the foramen magnum in the midsagittal plane. |
| **Nasion** | n | Intersection of the nasofrontal suture in the median plane |
| **Pronasale** | pn | The most anterior point of the nose. |
| **Prosthion** | pr | The most anterior point of the maxilla in the midsagittal plane. |

Two angles were also measured (na-ba-pn and n-ba-pr) to examine the position of prona-sale relative to the hard palate of the maxilla. The vertex of each triangle was positioned at basion with one ray to nasion. This was the starting point for both measurements. The second ray for each angle was positioned at pronasale for the first angle measurement and then to prosthion for the second angle measurement (Fig 1). Measurements were replicated seven days after initial assessment so that technical error of measurement (TEM), and relative TEM (rTEM) could be calculated from test-retest intra-observer measurements. The addition of another investigator to determine inter-observer errors was not included in this study as Stephan et al. [44] do not state that it is necessary to do both. Measurement errors for all variables assessed in this study were less than 0.3% for chimpanzee CT scans and less than 0.9% for human radiographs.

Descriptive statistics were presented for all measurements and their ratios. We then used simple linear regression to examine the relationship of nasal cavity length with cranial base length. We regressed ba-pn against ba-n and assessed intra- and interspecific regression slopes and intercepts using 95% confidence intervals. We further regressed ba-pr against ba-n to compare this relationship with that of the previous for both species, as well as the n-ba-pn angle against the n-ba-pr angle. Reduced major axis (RMA) regression was used to produce the predictive equations because RMA, unlike ordinary least squared regression, is not influenced by random variation of individual measurements around the regression line [45]. All statistical analyses were carried out with the Statistical Package for the Social Sciences (SPSS®) software, v. 26.0 for Mac (SPSS Inc, Chicago, II, USA).

To assess the reliability of the RMA prediction formulae, a set of out-of-group tests were performed. In total, the out-of-group test sample consisted of 16 individuals. The first test was performed on four subjects that belonged to the same genus as the training sample (conspecific), i.e., *Homo* (*n* = 2) and *Pan* (*n* = 2), and the second test, which functioned as an interspecies compatibility test (intraspecific), was performed on *Pan paniscus* (*n* = 1), *Gorilla gorilla* (*n* = 3), *Pongo pygmaeus* (*n* = 1), *Pongo abelli* (*n* = 1), *Symphalangus syndactylus* (*n* = 3), and *Papio hamadryas* (*n* = 3). Craniometric measurements were collected from each specimen and used with the appropriate regression model to predict pronasale position. All non-human primate subjects were collected from the Digital Morphology Museum, KUPRI (dmm.pri.kyoto-u.ac.jp), with the exception of the *Pan paniscus* subject (S9655) and the *Pan troglodytes* subject (S12652), which were downloaded from Morphosource (https://www.morphosource.org), as well as the *Pongo abelli* subject. The *Pongo abelli* subject was scanned as part of a health assessment using the Siemens Biograph mCT PET/CT system at the South Australian Health and Medical Research Institute (SAHMRI). Slice thicknesses were set at 0.6mm and the animal was sedated and positioned in the supine position during the scanning procedure. The CT scans were then donated by Zoos South Australia to the University of Adelaide for scientific research. A complete list of all out-of-group test material used in this study and their sources is presented in the S1 Table.

Exact ages for the infant *P. troglodytes* and *G. gorilla* are not provided by KUPRI, so we could only approximate their ages based on the dentition visible in their small immature jaws. In both *G. gorilla* and *P. troglodytes*, eruption of the first permanent molars occurs at approximately three years of age [46]. No permanent dentition eruption is visible in the *G. gorilla* subject, although the first permanent molars, canines, and incisors are approaching eruption. Therefore, PRI-7902 is not much less than 3 years of age. In the *P. troglodytes* subject, only the first permanent molars are fully erupted, so PRI-7895 is similarly approximated as 3 years of age.

In addition to the extant hominids, nasal protrusions were approximated for nine ancient hominid skulls using the RMA prediction formulae. Only hominid fossils with complete crania were selected. The crania included were two specimens representing the *Paranthropus* genus (KNM-WT 17000; *P. aethiopicus* and OH5; *P. boisei*), two specimens representing the *Australopithecus* genus, (Sts 5; *A. africanus* and MH1; *A sediba*), and five specimens representing the genus *Homo* (KNM-ER 1813; *H. habilis*, KNM-WT 15000; *H. ergaster/erectus*, LES1; *H. naledi*, Kabwe 1; *H. rhodesiensis/heidelbergensis*, and Amud 1; *H. neaderthalensis*/Neandertals). The soft tissue of each hominid was constructed in an oil-based modelling medium by GV using pegs anchored at basion to guide the shape of the nasal profiles and their underlying anatomy. A complete list of all fossil crania in this study and their sources is presented in the S1 Table.

## Results

Average cranial base length (ba-n) of chimpanzees was found to be only 3.58 mm (3.3%, z-score = 0.61) greater than that of modern humans (M = 107.86, SD = 5.83, *n* = 19 and M = 104.28, SD = 5.71, *n* = 72 respectively), though T-test results show they formally differ significantly, *p* = 0.02 (2 tail). Average nasal cavity length (ba-pn) of chimpanzees (M = 131.49, SD = 8.03, *n* = 19) is 10.76 mm (8.2%, z-score = 1.34) greater than that of modern humans (M = 120.73, SD = 6.87, *n* = 72), *p* < 0.001 (2 tail). This is approximately two times greater than the difference observed for cranial base length. Average jaw protrusion (ba-pr) of chimpanzees (M = 149.62, SD = 11.20, *n* = 19) is 51.58 mm (34.4%, z-score = 4.60) greater than that of modern humans (M = 98.14, SD = 6.11, *n* = 72), *p* < 0.001 (2 tail). Average n-ba-pr angle of chimpanzees (M = 35.23, SD = 3.46, *n* = 19) is 6.49 degrees (18.4%, z-score = 1.88) smaller than that of modern humans (M = 41.72, SD 2.87, *n* = 72), *p* < 0.001 (2 tail), as is average n-ba-pn angle of chimpanzees (M = 21.60, SD = 2.72, *n* = 19), which is 5.66 degrees (26.2%, z-score = 2.08) smaller than that of modern humans (M = 27.26, SD = 2.27, *n* = 72), *p* = 0.02 (2 tail).

The length of the nasal cavity (ba-pn) was, on average, equivalent to 122.1% and 115.8% of the length of the cranial base (ba-n) in chimpanzees and modern humans respectively, differing by 6.3%. Ratios of nasal cavity length (ba-pn) to jaw protrusion (ba-pr) were diametrically different by 44.4%. The ratio of mean jaw protrusion to cranial base length was 138.7% in chimpanzees and 94.3% in modern humans. In other words, chimpanzees were observed to have a mouth that protrudes past the nasal cavity, whereas modern humans were found to have a nasal cavity that protrudes past the mouth even though lengths of the nasal cavity in both species are relatively similar. These results and descriptive statistics for all measurements are illustrated in Fig 2 and Table 2 respectively.

Simple linear regressions revealed that nasal cavity length (ba-pn) was strongly and significantly correlated with cranial base length (ba-n) in both chimpanzees and modern humans (Table 3). In fact, the correlation coefficients obtained for each species were identical (*r* = 0.78; Table 3). Similarly, n-ba-pn and n-ba-pr angles were strongly correlated in both species with

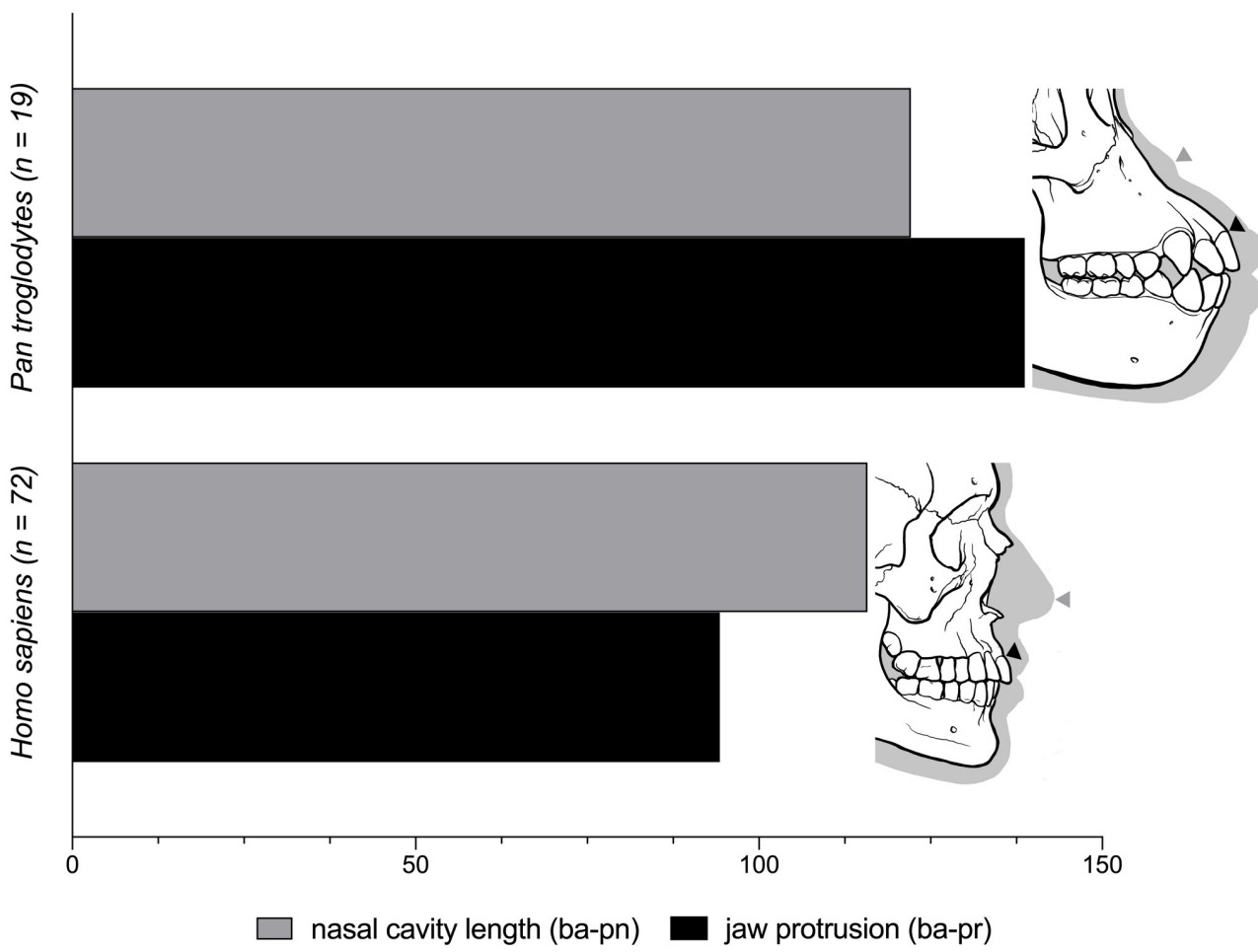

**Fig 2. Comparison of nasal cavity length (ba-pn) to jaw protrusion (ba-pr) in chimpanzees (*Pan troglodytes*) and modern humans (*Homo sapiens*).** See variable abbreviations in Table 1.

**Table 2. Descriptive statistics for cranial base length (ba-n), nasal cavity length (ba-pn), and jaw protrusion (ba-pr) in mm for chimpanzees (*Pan troglodytes*) and modern humans (*Homo sapiens*).** Angles measured in degrees are also shown.

| Variable[a] | Mean | SD | Minimum | Maximum |
|---|---|---|---|---|
| **Chimpanzee (*n* = 19)** | | | | |
| ba-n | 107.86 | 5.83 | 98.00 | 123.00 |
| ba-pn | 131.49 | 8.03 | 115.90 | 143.10 |
| ba-pr | 149.62 | 11.20 | 122.90 | 163.10 |
| n-ba-pr angle | 35.23 | 3.46 | 30.26 | 43.99 |
| n-ba-pn angle | 21.60 | 2.75 | 16.54 | 26.92 |
| **Human (*n* = 72)** | | | | |
| ba-n | 104.14 | 5.62 | 92.80 | 119.20 |
| ba-pn | 120.57 | 7.07 | 106.30 | 140.80 |
| ba-pr | 98.14 | 6.11 | 85.30 | 114.40 |
| n-ba-pr angle | 41.72 | 2.87 | 35.50 | 48.00 |
| n-ba-pn angle | 27.26 | 2.27 | 21.70 | 32.00 |

[a] See variable abbreviations in Table 1.

**Table 3. Ordinary least squares linear regressions of nasal cavity length (ba-pn) against cranial base length (ba-n) and n-ba-pr angle against n-ba-pn angle in chimpanzees and modern humans.**

| Variable[a] | R | Slope | Intercept | CI[b] Slope Lower | CI[b] Slope Higher | CI[b] Intercept Lower | CI[b] Intercept Higher |
|---|---|---|---|---|---|---|---|
| **Chimpanzee (*n* = 19)** | | | | | | | |
| ba-pn | 0.78* | 1.07 | 16.28 | 0.62 | 1.51 | -31.73 | 64.29 |
| n-ba-pn | 0.83 | 0.65 | -1.39 | 0.43 | 0.88 | -9.29 | 6.51 |
| **Human (*n* = 72)** | | | | | | | |
| ba-pn | 0.78* | 0.94 | 22.89 | 0.76 | 1.12 | 4.12 | 41.66 |
| n-ba-pn | 0.73 | 0.68 | -1.10 | 0.58 | 0.78 | -5.14 | 2.93 |

[a] See variable abbreviations in Table 1.

[b] 95% confidence interval.

* Indicates where correlations coefficients were identical between species.

chimpanzees (*r* = 0.83) having a slightly greater correlation coefficient than that of modern humans (*r* = 0.73; Table 3). Regression slopes were relatively consistent between species despite the differences mentioned previously regarding nasal cavity length and jaw protrusion. Regressions of nasal cavity length (ba-pn) against cranial base length (ba-n) had a positive slope of 1.07 for chimpanzees and 0.94 for modern humans. Regressions of n-ba-pn angle against n-ba-pr angle had a positive slope of 0.65 for chimpanzees and 0.68 for modern humans. These results show that slopes and intercepts are not species-specific. Subsequent regression analyses combining *Homo* and *Pan* data clearly show this consistency across the entire sample with each combined regression providing a slightly better fit for both species (Fig 3).

Given that simple linear regressions were able to identify statistically significant correlations in the combined sample, as well as establish homogeneity of interspecific covariation, we transformed the prediction equations using Reduced Major Axis (RMA) regressions to remove the influence of individual variation on the predictions. The position of the nasal tip for modern humans and chimpanzees could thus be approximated using the following equations:

$$\text{ba-pn (mm)} = 1.46 \, (\text{ba-n}) - 30.32 \pm 5.12$$

$$\text{n-ba-pn (degrees)} = 0.83 \, (\text{n-ba-pr}) - 7.42 \pm 1.30$$

The results of the out-of-group tests using the above RMA regression formulae are shown in Table 4. In the conspecific sample (i.e., *Homo sapiens* and *Pan troglodytes*), the average differences between observed and predicted ba-pn length and n-ba-pn angle was 2.3 mm and 0.8 degrees respectively. In the intraspecific sample (i.e., *Pan paniscus*, *Gorilla gorilla*, *Pongo pygmaeus*, *Pongo abelli*, *Symphalangus syndactylus*, and *Papio hamadryas*), a substantial difference in the predictive accuracy of the regression models among species was observed (Fig 4). For *P. paniscus* (*n* = 1) and *G. gorilla* (n = 3), the differences were rather small (mean difference for b-pn length = 1.4 mm; mean difference for na-ba-pn angle = 1.2 degrees). For *P. pygmaeus* (*n* = 1), *P. abelii* (*n* = 1), *S. syndactylus* (*n* = 3) and *P. hamadryas* (*n* = 3), the differences were larger (mean difference for b-pn length = 20.8 mm; mean difference for na-ba-pn angle = 2.8 degrees). This clearly indicates that there is an incremental decline in the predictive accuracy of the equations depending on the phylogenetic distance of these species relative to *H. sapiens* and *P. troglodytes*. Interestingly, the accuracy of angle predictions appeared to be affected less by phylogenetic position than length predictions.

It is important to emphasise the ages of several out-of-group test subjects. The ages of the two *Pan troglodytes* subjects was 3 and 5 years, the *P. paniscus* subject 4 years, the infant *G.*

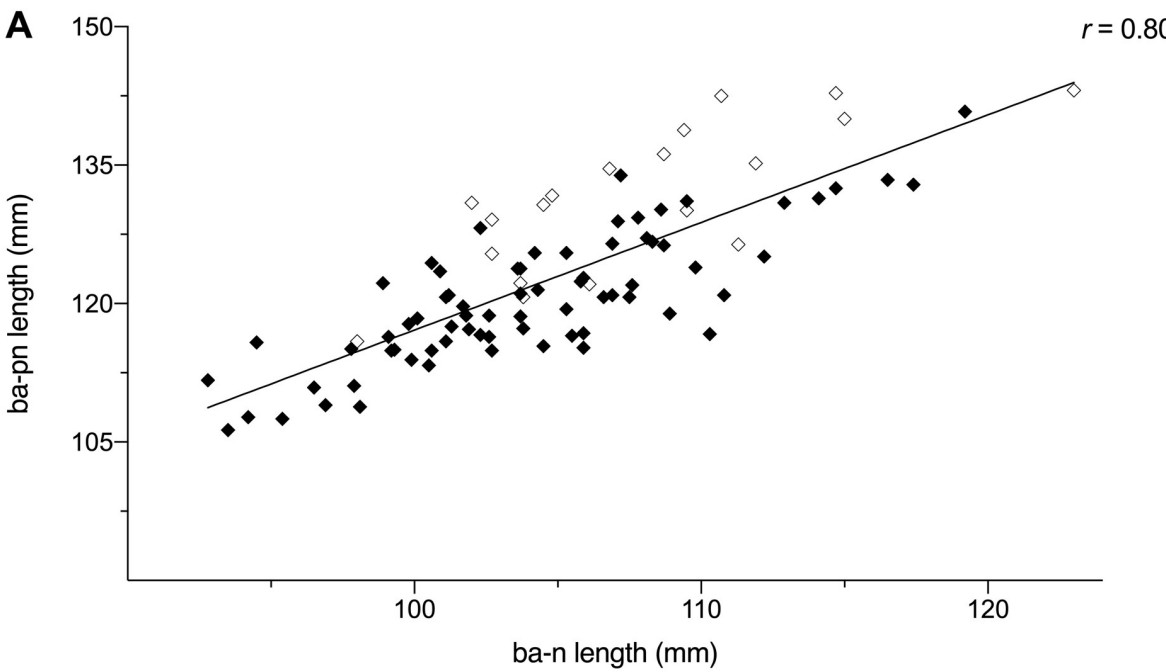

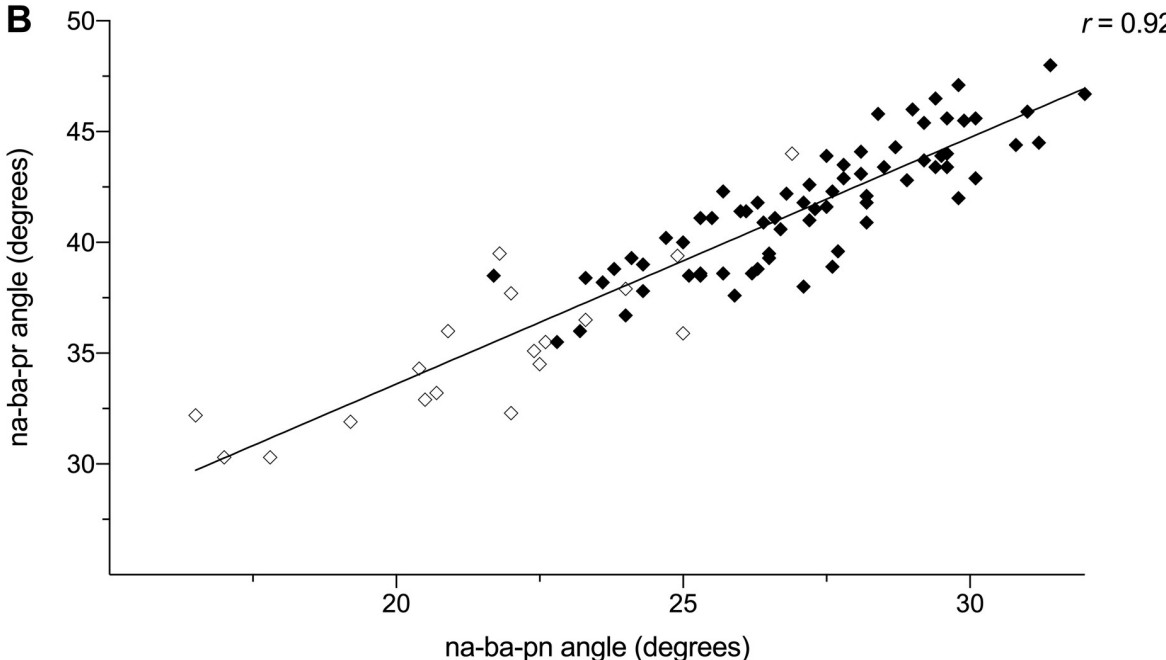

**Fig 3. Bivariate scatterplots showing regressions for a combined sample of chimpanzees (*Pan troglodytes*; ◇) and modern humans (*Homo sapiens*; ◆).** (A) Regression of nasal cavity length (ba-pn) on cranial base length (ba-n). (B) Regression of n-ba-pr angle on n-ba-pn angle. See variable abbreviations in Table 1.

**Table 4. Average differences between observed and predicted ba-pn length and n-ba-pn angle for subjects used in the out-of-group tests.** Subjects listed are grouped by species.

| Species | Length difference (mm) | Angle difference (degrees) | Length relative difference (%) | Angle relative difference (%) | n |
|---|---|---|---|---|---|
| **Conspecific sample (n = 4)** | | | | | |
| **H. sapiens** | 2.1 | 0.4 | 1.7 | 2.3 | 2 |
| **P. troglodytes** | 2.5 | 1.2 | 2.7 | 4.0 | 2 |
| **Intraspecific sample (n = 12)** | | | | | |
| **P. paniscus** | 1.1* | 0.6* | 1.0* | 2.0* | 1 |
| **G. gorilla** | 1.6 | 1.8 | 1.3 | 6.1 | 3 |
| **P. pygmaeus** | 10.8* | 0.2* | 8.7* | 1.1* | 1 |
| **P. abelli** | 15.4* | 1.2* | 12.8* | 4.5* | 1 |
| **S. syndactylus** | 17.4 | 4.3 | 21.3 | 22.4 | 3 |
| **P. hamadryas** | 39.4 | 5.5 | 38.8 | 16.7 | 3 |

* Indicates where values reported are for individuals only.

*gorilla* 3 years, the male *G. gorilla* 46 years, and the female *G. gorilla* 54 years. All of these subjects were therefore outside the age-range of the chimpanzee/human training sample. One of the human subjects in the out-of-group test was also outside the age-range of the training sample with an age of 3 years and 9 months. Given that the regression formulae were able to provide quite accurate estimates for all of these subjects (mean difference for b-pn length = 1.6 mm; mean difference for na-ba-pn angle = 1.3 degrees), it appears that when the regression

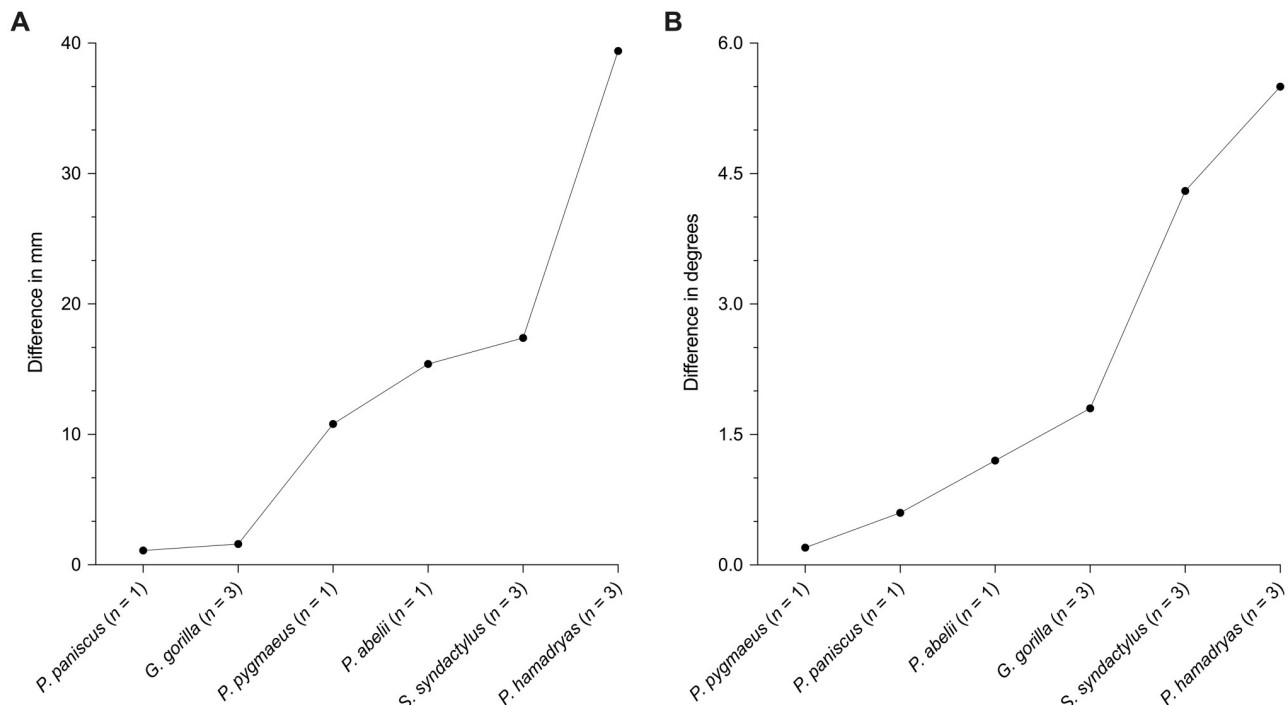

**Fig 4. Average difference between observed and predicted values shown for the out-of-group tests performed on six separate species that are outside of the chimpanzee/human training sample.** (A) Differences for nasal cavity length (ba-pn). (B) Differences for n-ba-pn angle. Notice the influence of the phylogenetic position of each species relative to modern humans and chimpanzees, i.e., from Hominoidea to Cercopithecoidea, and how this leads to a progressive increase in approximation error.

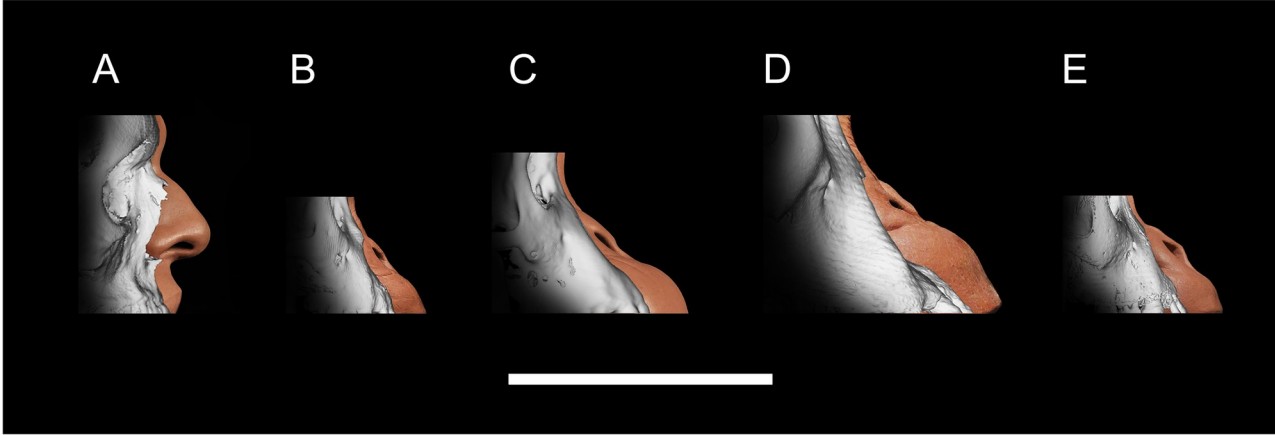

**Fig 5. Reduced major axis regression formulae applied in 3D approximations of the nasal region for out-of-group test subjects in norma lateralis.**
(A) *H. sapiens*: Anonymous 29-year-old male subject. (B) *P. troglodytes*: PRI-7895, 3-years-old. (C) *P. paniscus*: S9655, 4-years-old. (D) *G. gorilla*: PRI-Oki, 54-years-old. (E) *G. gorilla*: PRI-7902, 3-years-old. Scale Bar = 10 cm.

formulae are compatible with a given species, they do not appear to be restricted to a specific age-range.

The results of the regression formulae applied in 3D approximations of the nasal region for members of *H. sapiens*, *P. troglodytes*, *P. paniscus*, and *G. gorilla* are shown in Fig 5. The formulae of the present study have allowed for the objective and close approximation of the pronasale landmark for all of these species from measurements of their bone alone. 3D approximations were not performed for *P. pygmaeus*, *P. abelii*, *S. syndactylus*, and *P. hamadryas* because, as stated above, the prediction formulae produced poor estimates and are therefore incompatible with these species.

The results of the regression formulae applied in 3D approximations of the nasal regions for extinct hominids are presented in Fig 6. The results are consistent with previous interpretations of these species [7, 47]. There is significant variation in the nasal profile among hominid clades, from the chimp-like profiles of Pliocene *Australopithecus* to the human-like profiles of Pleistocene Neanderthals/Neandertals. Since we have observed that regression formulae derived from modern human and chimpanzee material demonstrate high reliability when applied to all African great apes, i.e., bonobos and gorillas, we put forward the hypothesis that the same formulae are applicable to ancient Plio-Pleistocene hominids and, since the approximations of nasal tip position were not produced using artistic intuition, that our results are empirically and scientifically accurate. The only caveat is that soft tissues of archaic hominids are nonexistent, thus making it impossible to assess the certainty of our results. This matter has been previously discussed in Montagu [48] and continues to burden the practice of Plio-Pleistocene hominid facial approximation today. Therefore, although the formulae allow one to empirically predict the position of pronasale, one must admit that the structures surrounding this landmark offer no certainty in their depictions. Therefore, we present them as informed hypotheses but nothing more.

## Discussion

Recently, Campbell et al. [38] showed facial soft tissue thicknesses covary with craniometric dimensions in chimpanzees, and that this allows for predictions of soft tissues from cranial measurements more precisely than by using averages of thicknesses. The present study identified another set of predictable relationships, this time for predicting pronasale position in a

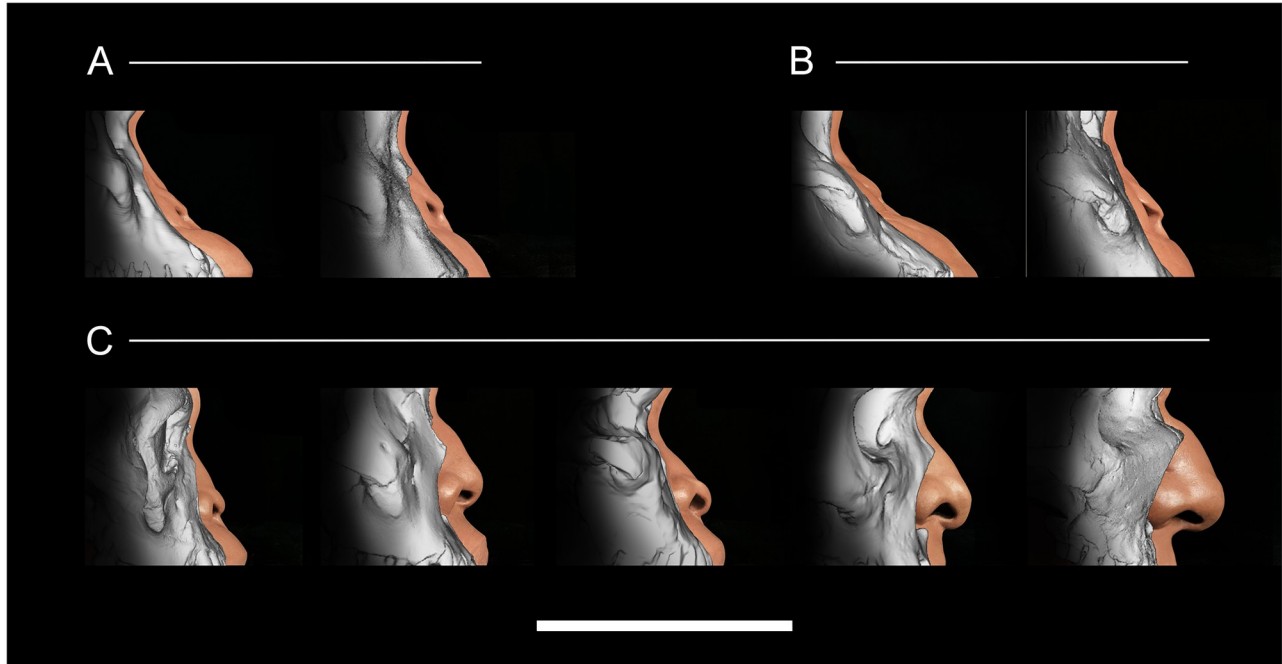

**Fig 6. Reduced major axis regression formulae applied in 3D approximations of the nasal region for extinct hominids in norma lateralis.** (A) *Australopithecus* genus: Sts 5 (*A. africanus*) and MH1 (*A. sediba*). (B) *Paranthropus* genus: KNM-WT 17000 (*P. aethiopicus*) and OH5 (*P. boisei*). (C) *Homo* genus: KNM-ER 1813 (*H. habilis*), KNM-WT 15000, (*H. ergaster / erectus*), LES1 (*H. naledi*), Kabwe 1 (*H. rhodesiensis/heidelbergensis*), and Amud 1 (*H. neaderthalensis*/Neandertals). Scale Bar = 10 cm.

similar way. Studies such as these are revealing that regression models actually outperform averages [18, 38, 49].

The interspecies out-of-group tests show the accuracy of our prediction models worsen with increasing taxonomic difference between humans/chimpanzees and other species. As can be seen in Fig 7, the regression trajectories of *P. paniscus* and *G. gorilla* appear to share a very close affinity with the chimpanzee/human training sample, whereas *P. pygmaeus*, *P. abelli*, *S. syndactylus*, and *P. hamadryas* appear to have group-specific slopes and intercepts that are different from our chimpanzee/human training sample. What is most interesting is that the RMA formulae used to predict pronasale position for the infant and two adult *G. gorilla* subjects produced accurate results, which not only highlights the validity of correlations identified in the present study but also, and more importantly, their compatibility with individuals of different ages belonging to separate species.

Based on the results of the out-of-group interspecies compatibility tests, we suggest that it can be formulated as a general rule that hominids with longer cranial base lengths tend to have longer nasal cavities, and that hominids with maxillae tilted further down from the Frankfurt horizontal plane tend to have axes of the nasal cavity also directed further downwards. Furthermore, since this has been identified in two extant species of hominid, which feature quite distinct skull morphologies, and that the regression models can reliably approximate pronasale position for other African great apes (i.e., *P. paniscus* and *G. gorilla*), these equations can be applied in facial approximation of extinct Plio-Pleistocene hominids. Quantitative linear regression essentially removes descriptive speculation during the approximation of this aspect of the nose for these species.

Owing to the relatively similar measurements of nasal cavity length among the individuals in our chimpanzee and modern human sample, our results are congruent with Nishimura

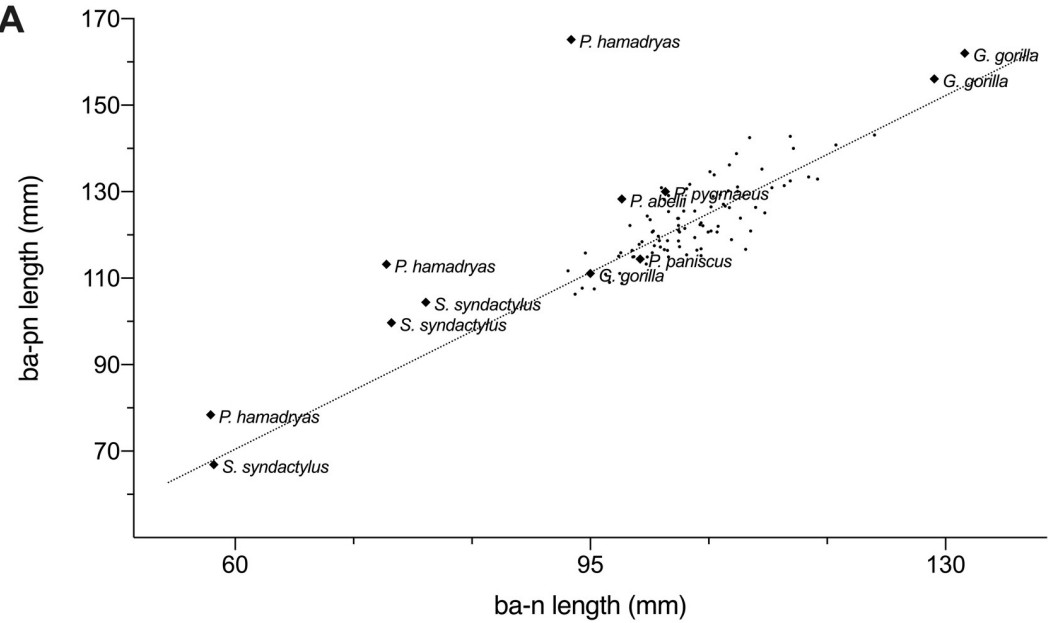

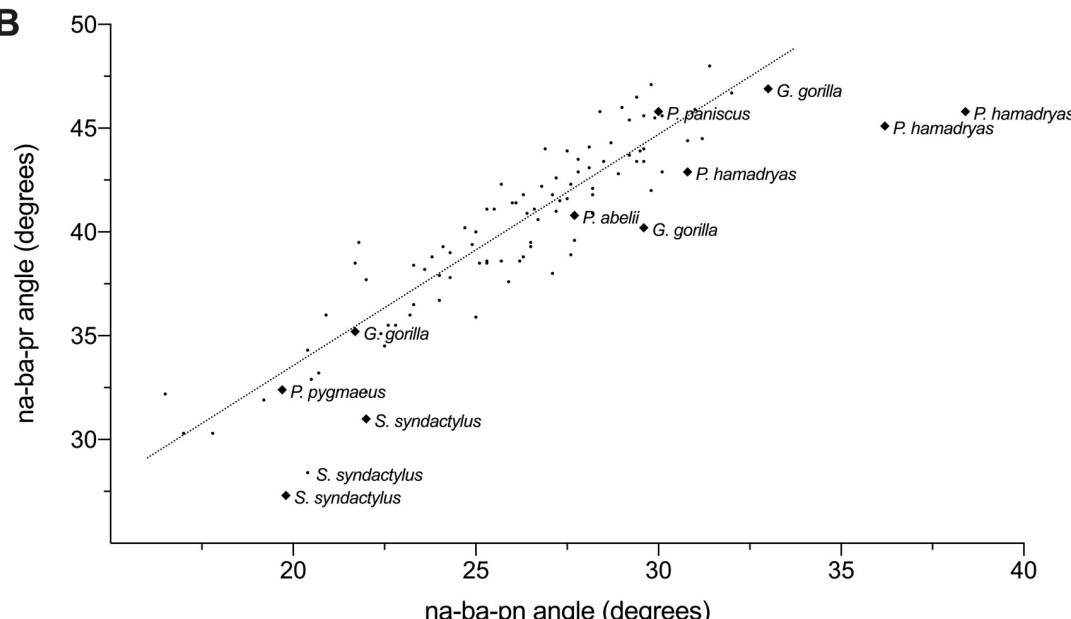

**Fig 7. Bivariate scatterplots with actual values for pronasale position in *Pan paniscus* (*n* = 1), *Gorilla gorilla* (*n* = 3), *Pongo pygmaeus* (*n* = 1), *Pongo abelli* (*n* = 1), *Symphalangus syndactylus* (*n* = 3), and *Papio hamadryas* (*n* = 3) superimposed over the chimpanzee/modern human regression lines.** (A) Regression of nasal cavity length (ba-pn) on cranial base length (ba-n). (B) Regression of n-ba-pr angle on n-ba-pn angle. See variable abbreviations in Table 1.

et al. [37] in that projecting noses are only partially a result of local adaptations to climate in the genus *Homo* [32]. Evolving humans entered a variety of habitats from tropical rainforest to open savannah, woodland mosaics, and eventually temperate climate niches [50]. Chimpanzees, on the other hand, did not migrate out of their ancestral environments. Despite this difference, nasal cavity lengths are similar in chimpanzees and modern humans. falsifying the hypothesis that local adaptations to climate included nasal cavity lengths in humans. It seems that nasal cavity length has been retained during human evolution while reductions of the masticatory apparatus reduced face size, thus revealing the prominent nose of modern humans. The nose did not protrude from the face, as previously hypothesized, the face shrunk around the nose exposing it over time. Unlike modern humans, chimpanzees did not evolve the same repertoire of extra-oral methods for predigesting food, nor did they change kinds of foods eaten, and, therefore, did not undergo any reductions in their masticatory apparatus. Furthermore, their social relationships based on male dominance did not allow canine reduction and loss of the C/P3 honing complex [51]. In contrast to changes in food preparation and canine use, neither of the species evolved any extra-nasal methods for conditioning inspired air, which is the most likely explanation for why dimensions of nasal cavity are so strikingly similar between modern humans and chimpanzees. Numerous clinically oriented studies have shown that there are at least two functions of the nose; humidification and temperature modification of inspired air [52, 53]. Nasal cavity length was clearly determined by natural selection for these adaptive roles, whose differences were minimal between great apes and humans relative to changes in the size of the masticatory apparatus.

Our analyses also concur with the observation that modern human facial growth is retarded relative to chimpanzees [54, 55]. This observation has been explored elsewhere in the neotenic theory of the human skull [56], and the self-domestication hypothesis [57]. It suffices to say that during great ape ontogeny, the facial prognathism increases from infancy to adulthood. In contrast, modern human skulls appear paedomorphic relative to chimpanzees. Thus, the developmental changes that occur throughout chimpanzee ontogeny, which result in a mouth that protrudes past the nose, do not occur in modern humans.

Our approximations of Plio-Pleistocene hominids shown in Fig 6 favor the hypothesis that the length of the nasal cavity remained relatively constant throughout human evolution. This point is made obvious by comparing the position of pronasale relative to prosthion between approximations in Fig 6. The more superior position of pronasale and anterior position of prosthion in relation to the piriform aperture, the more chimp-like the nose appears. In direct contrast, the more inferior position of pronasale and posterior position of prosthion, the more human-like the nose appears. Two hominids (KNM-ER 1813; *Homo habilis* and LES1; *Homo naledi*) appear to represent a morphology that is neither ape-like nor human-like. However, the approximations are not outside the realm of possibility of what could have been present in the morphology of intermediate species.

A limitation of this study is the low number of individuals in the chimpanzee sample and out-of-group tests. This is not uncommon for studies of great ape soft tissue due to their sparse availability. It is an unfortunate predicament that researchers find themselves in when studying the soft tissue parts of these endangered and protected animals. Osteological material is plentiful but soft tissue is relatively non-existent. As invaluable as the primate data from KUPRI are, there exists a need for the expansion of publicly available data to include a larger number of living, or recently deceased, individuals. Our solution to this problem is to collect existing data from primate sanctuaries and zoos since a large number of these animals have been scanned over the years for various health reasons (McLelland D. pers. communication). If we can make these data freely available, it would not only benefit researchers but also the public in providing an unprecedented look into the anatomy of our closest living

relatives. Such data could also be used to produce anatomically accurate, fully operable surgical training models for veterinarians and comparative anatomists alike. These are planned areas of research.

The other limitation of this study is that it only analyzed the length of the nose, which excludes other elements of the nasal form lateral to the mid-sagittal plane. Although pronasale position is the most defining point for the lateral view of the nose, other features of the nose are arguably as important as the protrusion alone. Alar and nostril size and shape have been somewhat investigated in modern humans but there is still a very wide-open field for researchers to describe these characters in non-human apes. To the knowledge of the authors, Hofer [30] is the only one to have formally studied and published descriptions of the soft tissue parts of the nose in nonhuman apes. In his analysis of gorilla, Hofer [30] recorded many interesting observations regarding their noses, including a number of sulci of the oro-nasal region that are not present in modern humans. The causes for these differences are unclear, so it is worth investigating this material further as explanations for these variations may provide clues to the possible presence of these features in ancient hominids. As such, this points to a large void in the literature requiring that other aspects of the nose receive equal attention in future studies.

Lastly, our modern human sample was composed of members of a USA population derived from Chinese and European backgrounds. It does, therefore, not necessarily reflect the true range of variation across geographically distinct human populations, such as samples of individuals that tend to be more prognathic. However, for the purpose of this study, it is not necessary to include all members of *Homo sapiens*. In the study of regression relationships, details about how prognathism affects the appearance of the nasal tip are provided by the chimpanzee sample. As our data illustrate, although chimpanzees are greatly more prognathic than modern humans, their nasal cavity lengths are relatively similar to those of humans. Therefore, other human populations, such as African populations, are likely to follow the same regression line and have very little, if any, effect on the present studies regression formulae.

## Conclusions

There are homogenous relationships between the skull and the soft tissue parts of the nose in both chimpanzees and modern humans as members of the hominid clade. Regressions combining chimpanzee and modern human measurements have shown that this amalgam of data can produce statistically reliable nasal tip location prediction formulae. These prediction formulae can be applied to chimpanzees, modern humans, bonobos, and gorillas. Based on these results, we hypothesize that the same formulae are valid for approximating the position of the nasal tip for extinct hominids because their crania fit into the range of variability present in our sample of extant hominid species. More investigations are needed to produce prediction formulae for other measurements of the soft parts of the nose, such as the ala nasi. Our study does not support the view that the nose arose out of the face. Rather, it provides evidence supporting the alternative hypothesis that reductions in the size of the masticatory apparatus over time led to the external appearance of the anterior part of the nasal cavity in the genus *Homo*.

## Supporting information

**S1 Table. List of non-human primate subjects included in this study.**
(DOCX)

## Acknowledgments

We would like to thank the Kyoto University Primate Research Institute (Kyoto, Japan) and Dr Ellie Simpson for facilitating the acquisition of the data used in this study. We would also like to thank Victor Surovec and Daniel Collins of Arizona State University for providing access to the Makerspace and the Vizproto Lab during the production of the approximations presented in Figs 5 and 6. We would like to acknowledge David McLelland from Zoos South Australia and Georgia Williams for facilitating the acquisition of Puspa's CT scan. The authors acknowledge the facilities, scientific and technical assistance of the National Imaging Facility, a National Collaborative Research Infrastructure Strategy (NCRIS) capability, at the Large Animal Research and Imaging Facility, South Australian Health and Medical Research Institute. We would also like to thank John Engelhardt for the illustration presented in Fig 1.

## Author Contributions

**Conceptualization:** Ryan M. Campbell, Gabriel Vinas, Maciej Henneberg.

**Data curation:** Ryan M. Campbell, Maciej Henneberg.

**Formal analysis:** Ryan M. Campbell, Gabriel Vinas, Maciej Henneberg.

**Investigation:** Ryan M. Campbell, Gabriel Vinas, Maciej Henneberg.

**Methodology:** Ryan M. Campbell, Gabriel Vinas, Maciej Henneberg.

**Project administration:** Ryan M. Campbell.

**Resources:** Ryan M. Campbell, Maciej Henneberg.

**Software:** Ryan M. Campbell.

**Supervision:** Maciej Henneberg.

**Validation:** Ryan M. Campbell, Maciej Henneberg.

**Visualization:** Ryan M. Campbell, Gabriel Vinas.

**Writing – original draft:** Ryan M. Campbell.

**Writing – review & editing:** Ryan M. Campbell, Gabriel Vinas, Maciej Henneberg.

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
