## [Decision Letter · Decision Letter 0]

15 Dec 2021

PONE-D-21-32794Relationships between the hard and soft dimensions of the nose in Pan troglodytes and Homo sapiens reveal the nasal protrusions of Plio-Pleistocene hominidsPLOS ONE

Dear Dr. Campbell,

Thank you for submitting your manuscript to PLOS ONE. After careful consideration, we feel that it has merit but does not fully meet PLOS ONE’s publication criteria as it currently stands. Therefore, we invite you to submit a revised version of the manuscript that addresses the points raised during the review process.

The reviewers comments are attached for resubmission with some revisions necessary before publication.

We look forward to receiving your revised manuscript.

Kind regards,

Caroline Wilkinson, PhD

Academic Editor

PLOS ONE

Reviewers' comments:

Reviewer's Responses to Questions

**Comments to the Author**

1. Is the manuscript technically sound, and do the data support the conclusions?

Reviewer #1: No

Reviewer #2: Partly

2. Has the statistical analysis been performed appropriately and rigorously? 

Reviewer #1: Yes

Reviewer #2: Yes

3. Have the authors made all data underlying the findings in their manuscript fully available?

Reviewer #1: Yes

Reviewer #2: Yes

4. Is the manuscript presented in an intelligible fashion and written in standard English?

Reviewer #1: Yes

Reviewer #2: Yes

5. Review Comments to the Author

Reviewer #1: Please see review attached. I have several major and some more minor concerns as outlined in the attachment.

The sample sizes are quite limited, and I think the authors are overinterpreting (and also overstating the value of) their results. The human sample does not encompass enough of human variation, needed to solve such a complex problem. I have a problem with the way in which the p-values were interpreted / stated. Please also see to the issue of ethics clearance.

Reviewer #2: This was well written and a very enjoyable read.

The authors analysed 19 CT scans of Chimpanzees heads and 72 lateral cephalometric radiographs of humans, and developed two equations to predict Nasal cavity length ba-pn and an facial angle na-ba-pr. The equations were tested on one human and one chimpanzee subject. The average difference between actual and predicted ba-pn length and na-ba-pn angle for the human subject was 1.9 mm and 1.4 mm for the chimpanzee subject (p value and statistical analysis unclear).

The authors then further tested the equation on 5 other species from various sources (n=12) and reported varying levels of accuracy for the individual species. The authors concluded that the accuracy was related to the phylogenetic distance of these species relative to Humans and Chimpanzees.

The equations were tested on different ages of non-human subjects between 3-54 year. The authors suggests the formulae are not restricted by age.

It was not entirely clear why those three specific measurements were chosen, please expand on how these measurements are relevant to nasal prediction and why were they selected. It was not clear whether the re-measurements were inter or intra observer, please clarify.

It will be helpful if a unit of measurements can be given for the two equations on line 313 and 314. I assumed its millimetres and degrees angle.

na-ba-pr, what is na? does na=n? please clarify

Sex of the subjects were documented, were they any difference between sex? if this was not conducted, why?

The accuracy of the out-of-group test was 1.9mm and 1.4mm for human and chimpanzee subjects. How does this number correlate with the na-ba-pn angle? I assumed the millimetre was related to the ba-pn length. Please expand and clarify.

The out-of-group test used one human and one chimpanzee. The authors reported that the results were 'quite accurate'. This is not a meaningful suggestion, more should be tested and a statistical analysis should be obtained. Please attempt to increase your test group.

In the discussion the authors discussed nasal cavity size, why was only the length measured and not the width? Surely the size of a cavity should include width and length as it is a 3D space.

Is figure 4 showing the difference for ba-pn in mm? What about the difference for the na-ba-pr angle? please clarify.

Is Figure 6 showing the noses that the author had reconstructed using the formulae? It will be more meaningful if the underlaying bone can be seen as a transparent overlay.

Overall, this research is novel and could be valuable to the field of palaeoanthropology, facial estimation, and developing an understanding on the relationship between the soft and hard tissues of hominid skulls.

6. PLOS authors have the option to publish the peer review history of their article (what does this mean?). If published, this will include your full peer review and any attached files.

Reviewer #1: No

Reviewer #2: No

---

## [Author Response · Author response to Decision Letter 0]

28 Jan 2022

BELOW WE RESPOND TO REVIEWERS’ COMMENTS INSERTING OUR RESPONSES IN CAPITALS (FOR EASY DISTINCTION) AFTER EACH COMMENT. WE ARE GRATEFUL TO REVIEWERS FOR THEIR HELPFUL COMMENTS, AND THE EDITOR FOR PROCESSING OUR MANUSCRIPT.

WE HAVE RESPONDED TO ALL 26 COMMENTS RAISED BY THE REVIEWERS, AND WE HAVE COMPLIED WITH 22 OF THEM. OUR RESPONSES REGARDING THE REMAINING 4 ARE PROVIDED BELOW.

 

Responses to Reviewer #1:

Comment 1: The results are only usable to very roughly predict the position of the tip of the nose. This should be acknowledged.

BASED ON OUR RESULTS, WE CANNOT SAY THE REGRESSION MODELS ARE ONLY USABLE TO “VERY ROUGHLY” PREDICT THE POSITION OF THE TIP OF THE NOSE. IN THE CONSPECIFIC SAMPLE (I.E., HOMO SAPIENS AND PAN TROGLODYTES; N = 4), THE AVERAGE DIFFERENCES BETWEEN OBSERVED AND PREDICTED BA-PN LENGTH AND N-BA-PN ANGLE WAS 2.3 MM AND 0.8 DEGREES RESPECTIVELY. IN THE INTRASPECIFIC SAMPLE (I.E., P. PANISCUS AND G. GORILLA; N = 4), THE AVERAGE DIFFERENCES BETWEEN OBSERVED AND PREDICTED BA-PN LENGTH AND N-BA-PN ANGLE WAS 1.4 MM AND 1.2 DEGREES RESPECTIVELY. IF THE REVIEWER MEANT TO SAY THAT OUR RESULTS ONLY ALLOW US TO PREDICT A SMALL PART OF THE OVERALL NASAL APPEARANCE, THIS IS CORRECT. HOWEVER, THIS IS ACKNOWLEDGED IN THE MANUSCRIPT, ESPECIALLY IN THE DISCUSSION. WITH THAT SAID, WE HAVE CHANGED THE TITLE OF THE MANUSCRIPT TO THE FOLLOWING: RELATIONSHIPS BETWEEN THE HARD AND SOFT DIMENSIONS OF THE NOSE IN PAN TROGLODYTES AND HOMO SAPIENS REVEAL THE POSITION OF THE NASAL TIP OF PLIO-PLEISTOCENE HOMINIDS. WE HAVE DONE THIS TO MAKE IT CLEAR FROM THE OUTSET THAT OUR RESULTS ARE ONLY RELEVANT TO THE NASAL TIP.

Comment 2: On the second page of the introduction, the authors list a few methods / references that have been used (in humans) to establish pronasale – this list is quite outdated and does not really illustrate the complexity of the problem. In this regard, see Carl N. Stephan, Jodi M. Cample, Pierre Guyomarc’h & Peter Claes (2019) An overview of the latest developments in facial imaging, Forensic Sciences Research, 4:1, 10-28, references 66 – 80 for a more recent view on this issue.

FOUR OF THE REFERENCES WERE ALREADY INCLUDED IN THE SECOND PAGE OF THE INTRODUCTION OF THE MANUSCRIPT. WE HAVE ADDED THE REFERENCE FOR CARL N. STEPHAN, JODI M. CAMPLE, PIERRE GUYOMARC’H & PETER CLAES (2019), AS WELL AS ALL OTHER REFERENCES THAT WERE RELEVANT TO PREDICTING THE NASAL PROFILE.

Comment 3: I found the human reference sample problematic. It included only Chinese and American/ European individuals and does not include any African samples. This should be corrected, as a sample of individuals who tend to be more prognathic would most certainly have strengthened the results and added more variation. I do not think the current sample adequately represents human variation.

IT IS NOT NECESSARY TO INCLUDE ALL GEOGRAPHICALLY DISTINCT POPULATIONS OF MODERN HUMANS IN OUR TRAINING SAMPLE. ALL INFORMATION ABOUT PROGNATHISM AND HOW IT AFFECTS THE APPEARANCE OF THE TIP OF THE NOSE IS PROVIDED BY THE CHIMPANZEE SAMPLE. FURTHERMORE, THE AIM OF THE STUDY WAS TO INVESTIGATE LINEAR REGRESSION RELATIONSHIPS BETWEEN CRANIOMETRIC VARIABLES, NOT AVERAGES OF THOSE VARIABLES AND, AS OUR DATA ILLUSTRATE, ALTHOUGH CHIMPANZEES ARE MORE GREATLY PROGNATHIC THAN MODERN HUMANS, THEIR RELATIONSHIPS ARE THE SAME. THEREFORE, OTHER HUMAN POPULATIONS, SUCH AS AFRICAN HUMANS FOR EXAMPLE, ARE LIKELY TO FOLLOW THE SAME REGRESSION LINE. IN ADDITION, WHILE OUR HUMAN SAMPLE DID NOT INCLUDE ANY AFRICAN MEMBERS, IT IS UNLIKELY THAT THEIR INCLUSION WOULD EFFECT THE ACCURACY OF THE REGRESSION MODELS FOR PREDICTING THE POSITION OF THE TIP OF THE NOSE. WE HAVE ADDED THIS INTO THE DISCUSSION.

Comment 4: I wondered about the issue of no ethics permission needed. The reference data being of humans would require some sort of permission. In our institution, an ethics waiver exists for data that are curated for research purposes. A committee oversees the granting this waiver to researchers, making sure that the data are used for ethical purposes. Some similar sort of permission from the gatekeepers of this database should be included.

THE NATIONAL STATEMENT ON ETHICAL CONDUCT IN HUMAN RESEARCH 2007 (UPDATED 2018) NOTES THAT (5.1.22): INSTITUTIONS MAY CHOOSE TO EXEMPT FROM ETHICAL REVIEW RESEARCH THAT:

(A) IS NEGLIGIBLE RISK RESEARCH (AS DEFINED IN PARAGRAPH 2.1.7); AND

(B) INVOLVES THE USE OF EXISTING COLLECTIONS OF DATA OR RECORDS THAT CONTAIN ONLY NON-IDENTIFIABLE DATA ABOUT HUMAN BEINGS.

THE PRESENT STUDY FALLS UNDER THIS EXEMPTION AND WE HAVE ATTACHED A LETTER FROM THE HUMAN RESEARCH ETHICS COMMITTEE AT THE UNIVERSITY OF ADELAIDE THAT CONFIRMS THIS.

Comment 5: The statement “It has been stated that primate comparative anatomy…” needs a reference.

THE APPROPRIATE REFERENCE HAS BEEN ADDED AFTER THIS STATEMENT.

Comment 6: “The aims of this study were to explore this matter further” – Clearly state the aim, i.e. which matter?

WE HAVE ADDED A SENTENCE CLARIFYING THE AIM OF THIS STUDY.

Comment 7: Related to the above, “second by performing a set of out-of-group tests” – Clearly state who are the out of group individuals, and why?

HERE, WE CAN ONLY ASSUME THAT THE REVIEWER HAS MISSED THE NEXT SENTENCE, WHICH CLEARLY STATES WHO THE OUT-OF-GROUP INDIVIDUALS WERE AND WHY THEY WERE INCLUDED. FOR CONVENIENCE, WE HAVE COPIED IT HERE EXACTLY AS IT WAS WRITTEN IN THE INITIAL SUBMISSION:

“THE FIRST TEST WAS PERFORMED ON TWO SUBJECTS THAT BELONGED TO THE SAME GENUS AS THE TRAINING SAMPLE, I.E., HOMO (N = 1) AND PAN (N = 1), AND THE SECOND TEST, WHICH FUNCTIONED AS AN INTERSPECIES COMPATIBILITY TEST, WAS PERFORMED ON PAN PANISCUS (N = 1), GORILLA GORILLA (N = 3), PONGO PYGMAEUS (N = 1), PONGO ABELLI (N = 1), SYMPHALANGUS SYNDACTYLUS (N = 3), AND PAPIO HAMADRYAS (N = 3).”

Comment 8: I am concerned about the inclusion of juveniles (p. 16). This has not been properly motivated and adds unnecessary complexity. This should be better motivated and discussed or else left out.

THE REALITY OF THE DREADFULLY LOW NUMBER OF READILY AVAILABLE CT DATA OF GREAT APES NECESSITATED THE USE OF AS MANY SUBJECTS AS POSSIBLE. WE DID NOT INTEND TO USE JUVENILES, BUT THE FACT THAT THEY WERE ON-HAND, MADE THEM ANOTHER CANDIDATE FOR FURTHER MODEL-RELIABILITY. THE FACT THAT THEY DO INDEED WORK ON ADULTS AND JUVENILES IS A TESTAMENT TO THEIR RELIABILITY AND ADDS FURTHER EVIDENCE FOR THE SUITABILITY FOR ALL AFRICAN GREAT APES. THIS WAS A WONDERFUL SURPRISE TO US THAT BOLSTERS OUR CONFIDENCE IN THE REGRESSION MODELS, TO DENIGRATE ITS INCLUSION AS NEEDLESS COMPLICATION IS A DISSERVICE TO THE FINDING.

Comment 9: The TEM assessments – were these inter- or intra-observer repeat assessments?

WE HAVE CLARIFIED THAT THE MEASUREMENTS WERE INTRA-OBSERVER ASSESSMENTS. WE HAVE ALSO ADDED A SENTENCE STATING THAT INTER-OBSERVER MEASUREMENTS WERE NOT INCLUDED BECAUSE STEPHAN ET AL. DID NOT APPEAR TO SUGGEST THAT WE WERE OBLIGED TO DO BOTH.

Comment 10: P. 9, bottom, the relationship between nasal size and cranial size examined – not really cranial size, which is much more than the few dimensions used here.

WE HAVE CHANGED ‘CRANIAL SIZE’ TO ‘CRANIAL LENGTH’.

Comment 11: P. 10 2nd para: To DEMONSTRATE the reliability…. Change demonstrate to assess, as we do not want to pre-empt the results (demonstrate), but rather want to assess whether there is a relationship.

WE HAVE CHANGED THE WORD DEMONSTRATE TO ASSESS ON PAGE 10 OF THE MANUSCRIPT.

Comment 12: Training sample – two subjects only. This is really quite a small dataset which limits the results of the study.

THE TRAINING SAMPLE CONSISTS OF 72 MODERN HUMANS AND 19 CHIMPANZEES. HERE, WE THINK THE REVIEWER MEANT TO SAY THAT OUR OUT-OF-GROUP TESTS CONSISTED OF ONLY TWO SUBJECTS. TO ADDRESS THIS, WE HAVE INCLUDED ANOTHER CHIMPANZEE AND ANOTHER HUMAN IN THE CONSPECIFIC OUT OF GROUP TEST. WE WOULD LIKE TO REMIND THE REVIEWER THAT THE OUT-OF-GROUP TESTS INCLUDE THOSE MEMBERS OF SPECIES OUTSIDE OF THE TRAINING SAMPLE. SO, THE SUM OF THE OUT-OF-GROUP SAMPLE IS NOW 16 INDIVIDUALS.

Comment 13: P. 11 – It is nice that the method was tested on hominids, but of course this is only a nicety in this context and does not really prove anything. It shows if realistic-looking reconstructions are possible using this method, and nothing more. This should be clearly stated.

WE HAVE ADDED A SENTENCE CLARIFYING THAT THE REGRESSION FORMULAE ONLY ALLOW ONE TO PRODUCE REALISTIC LOOKING APPROXIMATIONS BUT NOTHING MORE.

Comment 14: In the first part of the Results section, the p-values are stated. The matter in which this is stated should be relooked – for example it is state that “Average nasal cavity length of chimpanzees is SOMEWHAT GREATER than that of modern humans…”, but the p value is quoted as p<0.001. This is highly statistically significant. Also, the “Average na-ba-pr angle of chimpanzees is only slightly smaller than that of modern humans” but again the p-value is <0.001. Are you trying to convince us of your point?

WE HAVE EDITED THIS SECTION, REPLACING THE ADJECTIVES USED TO DESCRIBE THE DIFFERENCES WITH ABSOLUTE VALUES, PERCENTAGES, AND Z-SCORES.

Comment 15: In the next paragraph – “The rations (ratios) were rather similar…” – they are not really similar 122.1 Versus 115.8. Also, the other ratios quoted. These need to be put into perspective.

WE HAVE REWORDED THIS SECTION GIVING EXACT DIFFERENCES OF PERCENTAGE VALUE WHILE REMOVING VERBAL COMMENTS.

Comment 16: The discussion overinterprets the available results and should be shortened to include only what can be realistically said from these results.

WE HAVE REDUCED THE DISCUSSION AND CONCLUSIONS FROM 2124 WORDS TO 1681 WORDS (-443 WORDS, 25% LESS), MODIFYING THE TEXT TO INCLUDE ONLY WHAT CAN BE SAID FROM OUR RESULTS.

Responses to Reviewer #2:

Comment 1: It was not entirely clear why those three specific measurements were chosen, please expand on how these measurements are relevant to nasal prediction and why were they selected.

WE HAVE ADDED A SENTENCE CLARIFYING WHY THESE MEASUREMENTS ARE RELEVANT TO NASAL PREDICTION AND WHY THEY WERE SELECTED.

Comment 2: It was no clear whether the re-measurements were inter of intra observer, please clarify.

WE HAVE CLARIFIED THAT THE MEASUREMENTS WERE INTRA-OBSERVER ASSESSMENTS. WE HAVE ALSO ADDED A SENTENCE STATING THAT INTER-OBSERVER MEASUREMENTS WERE NOT INCLUDED BECAUSE STEPHAN ET AL. DID NOT APPEAR TO SUGGEST THAT WE WERE OBLIGED TO DO BOTH.

Comment 3: It will be helpful if a unit of measurement can be given for the two equations on line 313 and 314. I assumed its millimetres and degrees angle.

THE UNITS OF MEASUREMENT HAVE BEEN ADDED FOR THE TWO EQUATIONS ON LINE 313 AND 314.

Comment 4: na-ba-pr, what is na? does na=n? please clarify

THIS WAS A TYPOGRAPHICAL ERROR. NA SHOULD HAVE BEEN WRITTEN AS N. THIS HAS BEEN CORRECTED WHERE NECESSARY THROUGHOUT THE MANUSCRIPT.

Comment 5: Sex of the subjects were documented, were they any difference between sex? If this was not conducted, why?

WE DID NOT INVESTIGATE SEX DIFFERENCES FOR THE FOLLOWING REASONS: 1) SEX IS OFTEN SUBJECT TO DEBATE IN ARCHAIC HOMINIDS, SO THERE IS NO PRACTICAL REASON TO HAVE SEPARATE REGRESSION FORMULAE FOR MALES AND FEMALES WHEN THESE CANNOT BE ACCURATELY ASSIGNED TO HOMINIDS; AND 2) VARIATION IN HUMAN MORPHOLOGICAL CHARACTERISTICS IS FOUND MOSTLY IN INDIVIDUALS. ONLY 25% OF VARIATION IS A RESULT OF SEXUAL DIMORPHISM (SEE HENNEBERG, 1992). THE AIM OF OUR STUDY WAS TO INVESTIGATE LINEAR REGRESSION RELATIONSHIPS BETWEEN CRANIOMETRIC VARIABLES, NOT AVERAGES OF THOSE VARIABLES BETWEEN INDIVIDUALS CLASSIFIED BY SEX. WE HAVE ADDED THIS INTO THE DISCUSSION.

MACIEJ HENNEBERG, 1992. CONTINUING HUMAN EVOLUTION: BODIES, BRAINS, AND THE ROLE OF VARIABILITY. TRANSACTIONS OF THE ROYAL SOCIETY OF SOUTH AFRICA.

Comment 6: The accuracy of the out-of-group test was 1.9mm and 1.4mm for human and chimpanzee subjects. How does this number correlate with the na-ba-pn angle? I assumed the millimietre was related to the ba-pn length. Please expand and clarify.

THIS WAS A TYPOGRAPHICAL ERROR. THE AVERAGE DIFFERENCES HAVE NOW BEEN PRESENTED FOR THE LENGTHS AND ANGLES SEPARATELY IN THE TEXT OF THE MANSCURPT AS WELL AS IN A NEW TABLE (TABLE 4).

Comment 7: The out-of-group test used on human and one chimpanzee. The authors reported that the results were ‘quite accurate’. This is not a meaningful suggestion, more should be tested and a statistical analysis should be obtained. 

THIS ENTIRE SECTION HAS BEEN REWRITTEN. SIMPLE DIFFERENCES AND PERCENTAGE CHANGES HAVE BEEN CALCULATED AND PRESENTED FOR EACH SPECIES IN A NEW TABLE (TABLE 4). FIGURE 4 HAS ALSO BEEN REVISED TO SHOW THE DIFFERENCES FOR BA-PN LENGTH AND NA-BA-PR ANGLE SEPARATELY.

Comment 7.5: Please attempt to increase your test group.

WE HAVE INCLUDED ANOTHER CHIMPANZEE AND ANOTHER HUMAN IN THE CONSPECIFIC OUT OF GROUP TEST. WE WOULD LIKE TO REMIND THE REVIEWER THAT THE OUT-OF-GROUP TESTS INCLUDE THOSE MEMBERS OF SPECIES OUTSIDE OF THE TRAINING SAMPLE. SO, THE SUM OF THE OUT-OF-GROUP SAMPLE IS NOW 16 INDIVIDUALS.

Comment 8: In the discussion the authors discussed nasal cavity size, why was only the length measured and not the width? Surely the size of the cavity should include width and length as it is a 3D space.

WE DID NOT MEASURE ANY WIDTHS BECAUSE WE DID NOT MEASURE ANYTHING OUTSIDE OF THE MID-SAGITTAL PLANE. OUR PAPER IS ONLY CONCERNED WITH THE TIP OF THE NOSE AND WE ONLY HAD LATERAL CEPHALOGRAMS OF MODERN HUMANS. WE HAVE CHANGED THE WORD SIZE IN TO DISCUSSION TO LENGTH TO CLARIFY THIS. THE REVIEWER IS CORRECT IN THAT NASAL CAVITY SIZE WOULD TAKE INTO ACCOUNT WIDTH MEASUREMENTS, HOWEVER, OUR ANALYSIS IN REGARD TO LENGTH IS STILL VALID.

Comment 9: Is figure 4 showing the difference for ba-pn in mm? What about the difference for the na-ba-pr angle? Please clarify.

WE HAVE REVISED FIGURE 4 TO SHOW THE DIFFERENCES FOR BA-PN LENGTH AND NA-BA-PR ANGLE SEPARATELY.

Comment 10: Is Figure 6 showing the noses that the author had reconstructed using the formulae? It will be more meaningful if the underlying bone can be seen as a transparent overlay.

WE HAVE INSERTED THE UNDERLYING BONE AS AN OVERLAY TO ALL OF THE NOSES SHOWN IN FIGURES 5 & 6. WE AGREE THAT THE FIGURES ARE MORE MEANINGFUL WITH THIS ADDED.

---

## [Editor Report · Decision Letter 1]

2 Feb 2022

Relationships between the hard and soft dimensions of the nose in Pan troglodytes and Homo sapiens reveal the positions of the nasal tips of Plio-Pleistocene hominids

PONE-D-21-32794R1

Dear Dr. Campbell

We’re pleased to inform you that your manuscript has been judged scientifically suitable for publication and will be formally accepted for publication once it meets all outstanding technical requirements.

Kind regards,

Caroline Wilkinson, PhD

Academic Editor

PLOS ONE
---

## [Editor Report · Acceptance letter]

8 Feb 2022

PONE-D-21-32794R1 

Relationships between the hard and soft dimensions of the nose in *Pan troglodytes* and *Homo sapiens* reveal the positions of the nasal tips of Plio-Pleistocene hominids 

Dear Dr. Campbell:

I'm pleased to inform you that your manuscript has been deemed suitable for publication in PLOS ONE. Congratulations! Your manuscript is now with our production department. 

Kind regards, 

on behalf of

Professor Caroline Wilkinson 

Academic Editor

PLOS ONE